# Clinical Value of Whole Blood Procalcitonin Using Point of Care Testing, Quick Sequential Organ Failure Assessment Score, C-Reactive Protein and Lactate in Emergency Department Patients with Suspected Infection

**DOI:** 10.3390/jcm8060833

**Published:** 2019-06-12

**Authors:** Bo-Sun Shim, Young-Hoon Yoon, Jung-Youn Kim, Young-Duck Cho, Sung-Jun Park, Eu-Sun Lee, Sung-Hyuk Choi

**Affiliations:** Department of Emergency Medicine, Korea University College of Medicine, 08308 Seoul, Korea; ws4106@naver.com (B.-S.S.); yellowwizard@hanmail.net (J.-Y.K.); rionen@hanmail.net (Y.-D.C.); eusunlee@gmail.com (S.-J.P.); kuedpsj@daum.net (E.-S.L.); kuedchoi@korea.ac.kr (S.-H.C.)

**Keywords:** procalcitonin, point-of-care testing, bacteremia, mortality

## Abstract

We investigated the clinical value of whole blood procalcitonin using point of care testing, quick sequential organ failure assessment score, C-reactive protein and lactate in emergency department patients with suspected infection and assessed the accuracy of the whole blood procalcitonin test by point-of-care testing. Participants were randomly selected from emergency department patients who complained of a febrile sense, had suspected infection and underwent serum procalcitonin testing. Whole blood procalcitonin levels by point-of-care testing were compared with serum procalcitonin test results from the laboratory. Participants were divided into two groups—those with bacteremia and those without bacteremia. Sensitivity, specificity, positive predictive value, negative predictive value of procalcitonin, lactate and Quick Sepsis-related Organ Failure Assessment scores were investigated in each group. Area under receiving operating curve of C-reactive protein, lactate and procalcitonin for predicting bacteremia and 28-day mortality were also evaluated. Whole blood procalcitonin had an excellent correlation with serum procalcitonin. The negative predictive value of procalcitonin and lactate was over 90%. Area under receiving operating curve results proved whole blood procalcitonin to be fair in predicting bacteremia or 28-day mortality. In the emergency department, point-of-care testing of whole blood procalcitonin is as accurate as laboratory testing. Moreover, procalcitonin is a complementing test together with lactate for predicting 28-days mortality and bacteremia for patients with suspected infection.

## 1. Introduction

The systemic inflammatory response syndrome (SIRS) criteria comprising body temperature, heart rate, respiratory rate and white blood cell count, is used for sepsis diagnosis. However, it has been argued that the sensitivity and specificity of SIRS criteria are limited in accuracy for the purpose of screening. A new recommended test, quick sequential organ failure assessment (qSOFA) score, was introduced by the Sepsis-3 task force in 2016 [1]. qSOFA criteria consist of low blood pressure (systolic blood pressure ≤ 100 mmHg), increased respiratory rate (≥ 22 bpm) and altered mental status (Glasgow Coma Scale ≤ 14). qSOFA criteria show better specificity but lower sensitivity than SIRS criteria for predicting sepsis. Several trials were conducted to improve the performance of qSOFA as a screening tool. In a study, the addition of lactate level to qSOFA score performed better than qSOFA alone in emergency department (ED) patients with suspected sepsis [2].

Lactate is one of the useful parameters in sepsis. Although lactate was not included in the qSOFA model construction, the Sepsis-3 task force recommended serum lactate levels as a possible substitute for some qSOFA variables. However, a number of non-specific conditions can elevate serum lactate levels and hence, serum lactate levels only cannot be used to predict bacterial sepsis without clinical judgment.

Early diagnosis of bacteremia and antibiotic use in sepsis are important but challenging in the ED [3,4]. Confirmation of sepsis diagnosis needs blood cultures, which are only available after 12 to 48 h. This delay led to the development and evaluation of new tests for blood stream infections [5,6].

The procalcitonin (PCT) is an effective diagnostic marker of bacterial infection. Procalcitonin results can be available within an hour, and its usefulness is well acknowledged in the ED setting. A meta-analysis revealed that sensitivity and specificity of PCT ranges from 66% to 89% and 55% to 78%, respectively, for predicting bacteremia [7].

The primary object of this study was to investigate the usefulness of whole blood PCT levels determined by point-of-care testing (POCT) in patients with suspected infection visiting EDs and assess the accuracy of whole blood PCT tests for more rapid determination of PCT levels.

The secondary object of this study was to determine the usefulness of procalcitonin, C-reactive protein (CRP), qSOFA score and lactate in predicting bacteremia and 28-day mortality.

## 2. Materials and Methods

### 2.1. Study Design and Setting

This study was a prospective observational study conducted among patients admitted to the ED at Korea University Guro Hospital (KUGH). KUGH is a 1050-bed facility affiliated with a medical college. Annually, about 65,000 patients visit the ED of KUGH. Case report forms were completed for all patients who underwent a whole blood PCT test by POCT, prospectively.

This study protocol was reviewed and approved by the Institutional Review Board of Korea University Guro Hospital (approval No. 2018GR20012). Written informed consent was obtained from patients or family members.

### 2.2. Study Participants and Inclusion Criteria

The study period was from June to December 2018. Participants were included in the analysis if they had a febrile sense, suspected infection and underwent serum PCT testing based on the ED physician’s decision. We used systemic random sampling method. During the study period, we selected the first patient visiting an emergency room after 9 am and 9 pm among the patients with suspected infection and enrolled the patients if they underwent serum PCT testing. We excluded patients who were under 18 years, pregnant or had limited life expectancy due to chronic diseases. (Figure 1) The sample size was calculated as 194 by referring to Sample Size Calculators for Designing Clinical Research and a previous study [8,9].

### 2.3. Data Sampling

Gender, age, vital signs, mental status, infection source and laboratory findings, including serum PCT and whole blood PCT were recorded.

### 2.4. Outcome Measure

Blood culture results, qSOFA score, percentage of intensive care unit admission and 28-day mortality were investigated. Participants were also grouped into either the bacteremia or without bacteremia group. Bacteremia was defined as a blood culture positive test result. Sensitivity and specificity of the qSOFA score, serum lactate, serum PCT and whole blood PCT level for bacteremia were also assessed.

Whole blood PCT levels using POCT (HUBI-PCT^TM^ (Humasis, Anyang, Korea) used in this study) were compared with serum PCT levels of all participants. When blood for fever was sampled from participants, a remnant 0.5 mL of whole blood was collected by ED staff and tested per the manufacturer’s manual. The results of POCT were compared with the results of serum PCT measured with a cobas® 8000 modular analyser series (BRAHMS PCT-Q, Brahms, Germany) in the laboratory.

Area under receiving operating curve (AUROC) of CRP, serum lactate, serum PCT and whole blood PCT were evaluated to predict bacteremia and 28-day mortality.

### 2.5. Statistical Analysis

An independent sample *t*-test was used for normally distributed continuous variables examined by the Kolmogorov-Smirnov normality test, and the Mann–Whitney U test was used to compare continuous variables with a skewed distribution. A chi-square test was used for categorical variables. Agreement between serum PCT and whole blood PCT was estimated using the Bland–Altman plot and intraclass correlation coefficient by MedCalc software (MedCalc Software, Broekstraat, Belgium). 

AUROC curves of bacteremia and 28-day mortality were plotted for serum PCT, whole blood PCT, CRP, and serum lactate by SPSS Statistics (SPSS 20.0, IBM, Chicago, IL, USA). 

## 3. Results

### 3.1. Baseline Characteristics of the Study Population 

A total of 199 patients, with an equal ratio of males to females, participated in this study. Baseline characteristics of participants are shown in Table 1. The most common mode of transportation was self-transport by vehicle. Average body temperature was 37.7 ± 1.1 °C measured by an ear thermometer. One hundred and sixty-one patients (80.9%) showed an alert mentality. Most common infection sources were lung (28.1%), urinary tract (27.1%) and hepatobiliary-pancreas (13.1%).

### 3.2. Clinical Outcome of Enrolled Patients

One hundred and eighty-two, 128 and 112 patients were provided with blood culture, urine culture, and sputum culture, respectively. One hundred and thirty-one (65.8%) patients tested positive in one or more of the blood, sputum and urine culture tests. The most common source of positive results was the sputum test. qSOFA score >2 was recorded in 74 (37.2%) patients. Thirty-nine (19.3%) patients were admitted to the intensive care unit, and 20 (10.1%) patients died by the 28th hospital day (Table 2).

### 3.3. The Comparison of Diagnostic Values in qSOFA Score, Lactate, Serum Procalcitonin and Whole Blood Procalcitonin for Bacteremia

Table 3 shows sensitivity and specificity of the qSOFA score, serum lactate, serum PCT and whole blood PCT level for bacteremia. Procalcitonin greater than 0.5 ng mL^−1^ had better clinical performance than the qSOFA score. The clinical performance of whole blood PCT was equal to serum lactate. The negative predictive value of PCT and lactate was over 90%.

### 3.4. Agreement Between the Whole Blood Procalcitonin and the Serum Procalcitonin

Bland–Altman analysis plot and intraclass correlation are shown in Figure 2. Intraclass coefficient value was 0.9080 (95% CI, 0.8790–0.9303). The bias was 0.3 ng mL^−1^. The limits of agreement (bias ± 1.96 SD) ranged from −4.6 to 5.1 ng mL^−1^.

### 3.5. Area Under Receiving OperatingCurve of C-Reactive Protein, Lactate, Whole Blood Procalcitonin for Predicting Bacteremia and 28-Day Mortality

Figure 3 and Figure 4 show AUROC of CRP, lactate, serum PCT and whole blood PCT for predicting bacteremia and 28-day mortality. For bacteremia, AUROC of CRP, lactate, serum PCT, and whole blood PCT were 0.619 (95% CI, 0.523–0.715), 0.748 (95% CI, 0.669–0.826), 0.736 (95% CI, 0.646–0.826) and 0.736 (95% CI, 0.642–0.833), respectively. Area under receiving operating curve of CRP, lactate, serum PCT and whole blood PCT were 0.666 (95% CI, 0.561–0.770), 0.766 (95% CI, 0.656–0.876), 0.731 (95% CI, 0.619–0.842) and 0.694 (95%CI, 0.557–0.831) for 28-day mortality.

## 4. Discussion

Traditional screening tool for sepsis patients was the SIRS criteria developed in a 1991 consensus conference [10]. However, the concept of classical SIRS criteria did not meet the new sepsis definition, because SIRS criteria are not ideal for distinguishing between the host’s normal response to infection and when there is no response. For the quick identification of mortality risk in suspected septic patients outside ICU, the third International Consensus Definitions for Sepsis and Septic Shock (Sepsis-3) recently recommended the qSOFA as a simple tool [1]. Although there were different results in recent studies, the problem is that qSOFA scores have low sensitivity [11,12,13,14,15]. To overcome the low sensitivity of qSOFA scores, several researchers studied newer methods for predicting mortality in sepsis and had meaningful results [2,16].

There is another useful biomarker for a variety of infections, including sepsis. PCT, a precursor for calcitonin, was suggested by Moya et al. [17]. Procalcitonin is currently recognised for its usefulness in diagnosing infections. The use of PCT can provide guidance for deciding whether or not to administer antibiotics by excluding infections [18,19,20].

The ED is the main place where unpredictable situations occur frequently. Emergency physicians are needed to make quick decisions on a broad spectrum of illnesses and injuries. Thus, variable POCT are now widely used at the bedside. With the development in technology, the use of POCT has increased. If used in conjunction with electronic medical records, the results of the POCT can be instantly shared with the health care provider [21,22]. In the past, most of the POCT showed only positive and negative results, but recently, POCT showing quantitative values have been used and applied to get quick test results [23,24,25,26].

Procalcitonin results can be obtained within 20 min by using a POCT instrument obtained immediately at the bedside; they help plan for prompt treatment options, such as antibiotics. Moreover, as indicated above, the qSOFA score has higher specificity than SIRS criteria in the 28-day mortality prediction but has lower sensitivity. For the prediction of bacteremia, whole blood PCT level was better in sensitivity than qSOFA score >2 in this study. In particular, the negative predictive value of whole blood PCT for predicting bacteremia was 90.0 (95% CI, 83.81–93.99). Considering the performance of PCT in this study, PCT level could be a reasonable reference to rule out bacteremia rather than to predict blood stream infection.

As blood is allowed to clot and the clot removed before the analysis, serum PCT test result takes more than half an hour, at least. Whole blood sample is usually preferred for quick results, especially in POCT. We compared the PCT level in both whole blood and plasma for consistency (Intraclass correlation coefficient = 0.9080). Given that quick results can be provided by POCT, our results showed the usefulness of PCT when whole blood samples were used.

Lactate has been established as an objective marker reflecting tissue perfusion, especially in sepsis. An increasing level of lactate is associated with mortality in sepsis [27,28,29]. In addition, AUROC of CRP, lactate, serum PCT and whole blood PCT were assessed. Serum PCT and whole blood PCT were superior to CRP and were similar to serum lactate in predicting bacteremia and 28-day mortality in sepsis. 

There are some limitations to this study. This study was limited to one hospital, with a random selection of a small sample size, which may pose an issue of faulty generalisation. To prove the advantage of POCT as a quick test, the actual time from blood sampling to getting the result of POCT should have been measured. According to the manufacturer’s manual, only 15 min are needed to obtain PCT results. We also did not consider other factors, such as how long it takes to collect blood and how long it takes to handle a POCT machine. Lastly, the long-term outcomes of participants were not recorded, as only bacteremia and 28-day mortality were of interest as an end point in this study.

## 5. Conclusions

In the emergency department, point-of-care testing of whole blood procalcitonin is as accurate as laboratory testing. Moreover, PCT is a complementing test together with lactate for predicting 28-days mortality and bacteremia for patients with suspected infection.

## Figures and Tables

**Figure 1 jcm-08-00833-f001:**
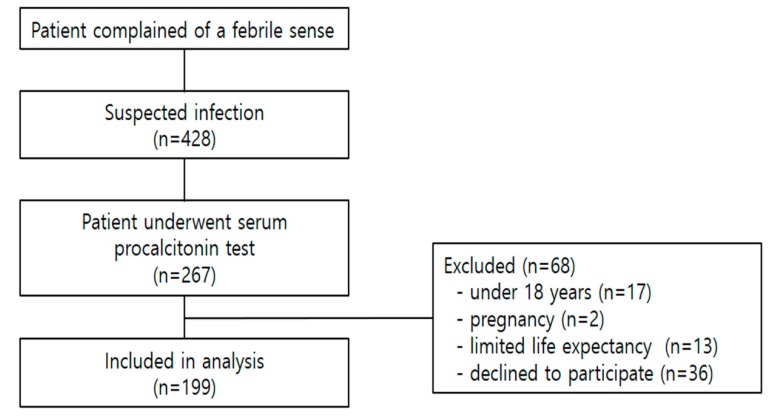
Flow chart of patient recruitment.

**Figure 2 jcm-08-00833-f002:**
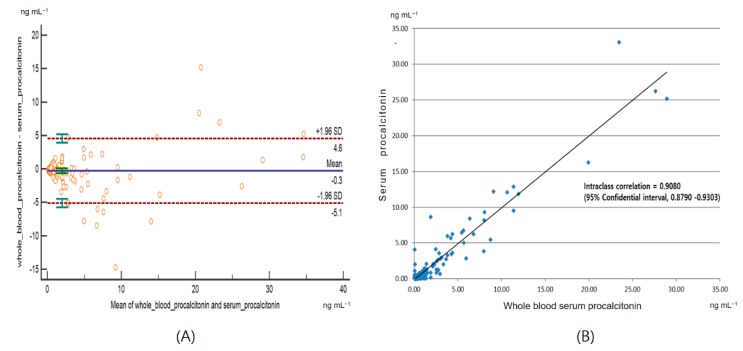
Bland–Altman plot (**A**) and intraclass correlation (**B**) between serum procalcitonin and whole blood procalcitonin.

**Figure 3 jcm-08-00833-f003:**
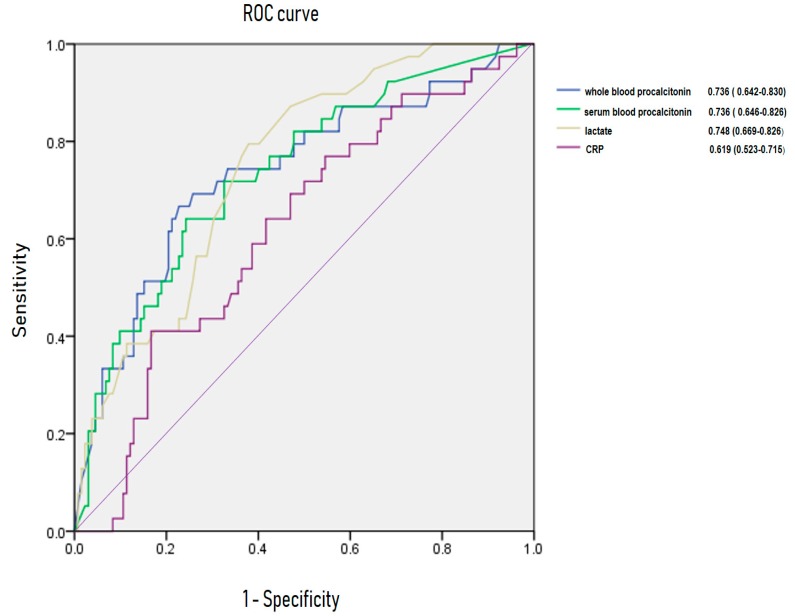
Area under receiving operating curve (AUROC) of C-reactive protein, lactate, procalcitonin for predicting bacteremia.

**Figure 4 jcm-08-00833-f004:**
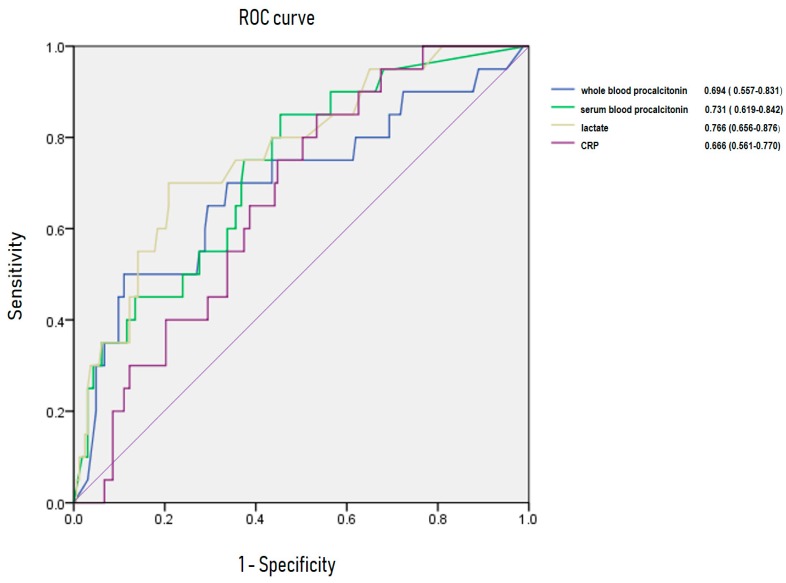
Area under receiving operating curve (AUROC) of C-reactive protein, lactate, procalcitonin for predicting 28-day mortality.

**Table 1 jcm-08-00833-t001:** Baseline characteristics of study participants.

Baseline Characteristic	Value
Age (year, mean ± standard deviation)	67.1 ± 17.1
Male (n, %)	100 (50.3)
Mode of transportation (n, %)	
Emergency medical service	52 (17.4)
Self-transport by vehicle	94 (47.2)
Transferred from another hospital	53 (26.6)
Vital sign (mean ± standard deviation)	
Systolic blood pressure (mmHg)	118.5 ± 28.6
Diastolic blood pressure (mmHg)	72.9 ± 16.8
Heart rate (beats min^−1^)	98.0 ± 22.1
Respiratory rate (breaths min^−1^)	21.9 ± 4.8
Body temperature (°C)	37.7 ± 1.1
Mental status (n, %)	
Alert	161 (80.9)
Verbal	14 (7.0)
Pain	23 (11.6)
Unresponsive	1 (0.5)
Infection source (n, %)	
Lung	56 (28.1)
Urinary tract	54 (27.1)
Gastrointestinal tract	16 (8.0)
Hepatobiliary-pancreas	26 (13.1)
Bone-joint-soft tissue	15 (7.5)
Central nervous system	6 (3.0)
Indwelling catheter	1 (0.5)
Endocarditis	1 (0.5)
Blood stream	10 (5.0)
Unknown	16 (8.0)
Laboratory finding	
White blood cell (×10^3^ μL^−1^)	11.4 ± 7.6
Platelet (×10^3^ μL^−1^)	197.7 ± 96.3
Creatinine (mg dL^−1^)	1.3 ± 1.4
Lactate (1.4 mmol L^−1^)	2.6 ± 2.2
C-reactive protein (mg L^−1^)	102.4 ± 96.7
Serum procalcitonin (ng mL^−1^)	2.9 ± 7.4
Whole blood procalcitonin (ng mL^−1^)	2.0 ± 4.2

**Table 2 jcm-08-00833-t002:** Clinical outcome of study participants.

**Positive result on the culture test**	**N/total N (%)**
Blood	41/182 (22.5)
Sputum	108/127 (85.0)
Urine	47/112 (42.0)
**qSOFA score**	**N (%)**
0	114 (57.3)
1	11 (5.5)
2	57 (28.6)
3	17 (8.5)
**Intensive care unit admission (N, %)**	39 (19.6)
**28 day mortality (N, %)**	20 (10.1)

qSOFA, Quick Sepsis-related Organ Failure Assessment.

**Table 3 jcm-08-00833-t003:** Sensitivity and specificity of the qSOFA score, serum lactate and whole blood procalcitonin level for bacteremia.

Positive Blood Culture	qSOFA Score ≥2 (95%CI)	Whole Blood Procalcitonin >0.5 ng mL^−1^ (95%CI)	Serum Blood Procalcitonin >0.5 ng mL^−1^ (95%CI)	Serum Lactate >2 mmol L^−1^ (95%CI)
Sensitivity	34.15 (20.08–50.59)	75.61 (59.70–87.64)	82.93 (67.94–92.85)	79.49 (63.54–90.70)
Specificity	65.25 (56.78–73.06)	63.83 (55.32–71.75)	46.10 (37.68–54.69)	61.94 (53.16–70.18)
PPV	22.22 (15.00–31.62)	37.80 (31.48–44.57)	30.91 (26.68–35.48)	37.80 (31.73–44.29)
NPV	77.31 (72.61–81.41)	90.00 (83.81–93.99)	90.28 (82.21–94.91)	91.21 (84.65–95.13)
	*p* > 0.05	*p* = 0.000	*p* = 0.001	*p* = 0.000

qSOFA, Quick Sepsis-related Organ Failure Assessment; CI, confidence interval; PPV, positive predictive value; NPV, negative predictive value.

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
