# Peer review of "Clinical Value of Whole Blood Procalcitonin Using Point of Care Testing, Quick Sequential Organ Failure Assessment Score, C-Reactive Protein and Lactate in Emergency Department Patients with Suspected Infection"

_jcm, 2019, doi:10.3390/jcm8060833_

Reviewer 1 Report

The manuscript “The Clinical Usefulness of Whole Blood Procalcitonin Using Point of Care Testing for Bacterial Sepsis in the Emergency Department” illustrated a prospective study to compare the prognostic accuracy of serum PCT, Whole blood PCT (POCT) , lactate and qSOFA to predict bacteremia and 28-day mortality who presented in one of the urban hospital in Korea. These are the comments for the author:

Abstract

The statement “Area under receiving operative curve results proved whole blood procalcitonin to be useful…” may be over claimed. The AUROC for whole blood PCT is only 0.69 which is only fair in predicting 28-day mortality. 

The statement “determining whole blood procalcitonin can be a useful test for sepsis patients” is inaccurate. The AUROC for PCT 0.74 indicate PCT is fairly accurate. It maybe useful to predict bacteremia rather then sepsis. Sepsis is not bacteremia. Bacteremia is just a subset of sepsis.

Title: The title do not reflect the study objectives completely. The study did evaluate CRP and lactate  to predict bacteremia and the 28-day mortality.

Introduction:

The qSOFA is introduced as a parsimonious tool to identity sepsis related organ failure but do not replace SIRS as a screening tool. The objectives "to determine usefulness of CRP, qSOFA and lactate in predicting bacteremia and 28-day mortality" were missing.

Methodology:

Author shall elaborate the method of randomization in detail. 

Since this is a prospective study, estimated sample size should be calculated and mentioned in the manuscript.

How infections were suspected in ED? What criteria was used to determine it’s presence?

What is the study definition for sepsis?

The term 'disabled' and 'ill terminally' were rather non-specific. An appropriate definition needed  for these two terms.

Is that only blood C&S were traced to determine bacteremia? Were other cultures results were traced too and used to determine bacteremia? Noted In table 2 that other culture (sputum and urine) results were traced as well but not sure whether the results were used.

Ethical statement: Did the informed consent were obtained during the recruitment? This needs to be stated in the manuscript.

Results

Did all 199 subjects had their blood C&S taken? 

It will be great authors could provide a patient recruitment flow chart to clearly explain the study recruitment flow.

Line 104: The table’s title is incomplete. Table 3 showed qSOFA, PCT and lactate results but in the title, lactate and the outcome were missing.

Line 116: What  outcome was these AUROC results referring to? 

Discussion

Overall, AUROC for whole blood PCT in predicting bacteremia and 28-day mortality are fair even though it is strongly correlated with serum PCT result. Authors shall highlight what are the advantages of having bedside POCT PCT in predicting bacteremia as compare to laboratory serum POCT. Lactate and PCT have a similar fair prognostic value in predicting 28-days mortality. Author shall suggest PCT as a complementing test together with qSOFA for predicting 28-days mortality. 

Conclusion for the manuscript is missing.

Author Response

Dear Editor-in-Chief and reviewers

We appreciate the reviewers’ comments, which have improved our manuscript. We have addressed and responded to each reviewer comment in a point-by-point fashion as belows.

Revised phrases were colored as red in the manuscript.

Reviewer pointed out that moderate English changes are required. I have received English editing service in my institution. However, if you still think more English editing is needed, I want to do it through MDPI English editing services. 

Abstract

The statement “Area under receiving operative curve results proved whole blood procalcitonin to be useful…” may be over claimed. The AUROC for whole blood PCT is only 0.69 which is only fair in predicting 28-day mortality.

I changed the word “useful” into statistical wording “fair”. (line: 32)

The statement “determining whole blood procalcitonin can be a useful test for sepsis patients” is inaccurate. The AUROC for PCT 0.74 indicate PCT is fairly accurate. It maybe useful to predict bacteremia rather then sepsis. Sepsis is not bacteremia. Bacteremia is just a subset of sepsis.

I revised the phrase as follows

-  Moreover, PCT is a complementing test together with lactate for predicting 28-days mortality and bacteremia for patients with suspected infection. (line 33-35)

Title: The title do not reflect the study objectives completely. The study did evaluate CRP and lactate  to predict bacteremia and the 28-day mortality.

I revised the title as ‘Clinical value of whole blood procalcitonin using point of care testing, quick Sequential Organ Failure Assessment score, C-reactive protein and lactate in emergency department patients with suspected infection’ to express the objectives and results of the study properly. (line: 2-6)

Introduction:

The qSOFA is introduced as a parsimonious tool to identity sepsis related organ failure but do not replace SIRS as a screening tool. The objectives "to determine usefulness of CRP, qSOFA and lactate in predicting bacteremia and 28-day mortality" were missing.

I revised the introduction as “The primary object of this study sought was to investigate the usefulness of whole blood PCT levels determined by point-of-care testing (POCT) in sepsis patients with suspected infection visiting EDs, and assess the accuracy of whole blood PCT tests for more rapid determination of PCT levels. The secondary object of this study was to determine usefulness of procalcitonin, C-reactive protein (CRP), qSOFA score and lactate in predicting bacteremia and 28-day mortality.” (line:62-66)

Methodology:

Author shall elaborate the method of randomization in detail.

I added the sampling method in detail and revised the paragraph of Study Participants and Inclusion Criteria as follows

Participants were included in the analysis if they had a febrile sense, suspected infection and underwent serum PCT testing based on the ED physician decision. We used systemic random sampling method. During study period, we selected first patient visiting emergency room after 9 am and 9 pm among the patients with suspected infection, and enrolled the patients if they underwent serum PCT testing. We excluded patients who were under 18 years, pregnant or had limited life expectancy due to chronic diseases. (Figure 1) (line: 74-79)

Since this is a prospective study, estimated sample size should be calculated and mentioned in the manuscript.

I added a comment for sample size as follows: “Sample size was calculated as 194 by referring to sample Size Calculators for Designing Clinical Research (http://www.sample-size.net/correlation-sample-size/) and previous study [8].”

(line 80-81)

How infections were suspected in ED? What criteria was used to determine it’s presence?

Suspected infection in patient complained of a febrile sense was depend on the ED physician’s decision

I revised text as “Participants were included in the analysis if they had a febrile sense, suspected infection and underwent serum PCT testing based on the ED physician decision.” (line: 74-76)

What is the study definition for sepsis?

Strictly speaking, the subject of this study is not a patients corresponding to the definition of sepsis but only a suspected one. So, I modified the expression of “sepsis” as “suspected infection” and revised the whole manuscript in accordance with the meaning of the changed word.

The term 'disabled' and 'ill terminally' were rather non-specific. An appropriate definition needed  for these two terms.

For the sake of clarity, I revised the text as “We excluded patients who were under 18 years, pregnant or had limited life expectancy due to chronic diseases” (line: 789-79)

Is that only blood C&S were traced to determine bacteremia? Were other cultures results were traced too and used to determine bacteremia? Noted In table 2 that other culture (sputum and urine) results were traced as well but not sure whether the results were used.

We defined the bacteremia as positive peripheral blood cultures in a patient with signs and symptoms of infection per the textbook definition.  To avoid confusion, I added a definition of bacteremia in this study. (line: 89)

Ethical statement: Did the informed consent were obtained during the recruitment? This needs to be stated in the manuscript.

We obtained informed consent from all recruited patients. I added the phrase in ethics statement. (line 109-110)

Results 

Did all 199 subjects had their blood C&S taken?

One hundred and eighty two, 128 and 112 patients were provided with blood culture, urine culture and sputum culture, respectively. (line 119-120)

It will be great authors could provide a patient recruitment flow chart to clearly explain the study recruitment flow.

I added a Flow chart of patient recruitment in figure 1.

Line 104: The table’s title is incomplete. Table 3 showed qSOFA, PCT and lactate results but in the title, lactate and the outcome were missing.

I revised the table’s title. (line 125)

Line 116: What outcome was these AUROC results referring to?

I revised the phrase as follows

AUROC of CRP, lactate, serum PCT and whole blood PCT were 0.666 (95% CI, 0.561-0.770), 0.766 (95% CI, 0.656-0.876), 0.731 (95% CI, 0.619-0.842) and 0.694 (95%CI, 0.557-0.831) for 28-day mortality. (line 141)

Discussion

Overall, AUROC for whole blood PCT in predicting bacteremia and 28-day mortality are fair even though it is strongly correlated with serum PCT result. Authors shall highlight what are the advantages of having bedside POCT PCT in predicting bacteremia as compare to laboratory serum POCT. Lactate and PCT have a similar fair prognostic value in predicting 28-days mortality. Author shall suggest PCT as a complementing test together with qSOFA for predicting 28-days mortality.

I highlighted the advantages of having bedside POCT PCT in the discussion (line 171-175)

Conclusion was revised for PCT as a complementing test per your suggestion (line 191-193)

Conclusion for the manuscript is missing.

I added conclusion.

Reviewer 2 Report

Major Comments:

The selection of patients is unclear and potentially biased. Firstly, what is meant by "randomly selected"? What was the selection process? Why not taking all patients in a limited time frame? Secondly, selection from patients "...suspected infection, and underwent serum PCT testing…", what does this mean?, What were the criteria for testing serum PCT? In the reviwer's view, there was an uncontrolled patient selection bias.

It is mentioned that the standard serum PCT test results are available within one hour, which is quite reasonable. Hence, what is the real time that was saved by the new whole blood test? Did this faster test result in any quicker decisions? Were the whole blood PCT tests blinded to the ED physicians?

It remains unclear how the patients "without bacteremia" were defined. Were these 158 patients (199 total minus 41 BC positive?), although there may be another severe infection?? If so, what is the value of the high NPV? Not giving antibiotics to patients with severe pneumonia, but negative BC??

Linear regression is not the standard method to analyze a new test (whole blood PCT)  compared with a gold standard (serum PCT). Please provide a Bland-Altman analysis plus plot!!

It is absolutely unclear why in Table 3 and Figures 2 and 3 (AUROC), the standard serum PCT is left out?? Direct comparison is one of the key questions!

Minor Comments:

Abstract: What is an "accurate correlation"??? This is not a standard wording in statistics.

References: Some of the papers referenced (especially in the Intro) are more than 20 years old.

Statistics: How did the Authors test normal distribution?

Page 3, ll. 115 to 117: Probably the AUROC for mortality is meant. Please add.

Author Response

Dear Editor-in-Chief and reviewers

We appreciate the reviewers’ comments, which have improved our manuscript. We have addressed and responded to each reviewer comment in a point-by-point fashion as belows.

Revised phrases were colored as red in the manuscript.

Reviewer pointed out that moderate English changes are required. I have received English editing service in my institution. However, if you still think more English editing is needed, I want to do it through MDPI English editing services.

Major Comments:

The selection of patients is unclear and potentially biased. Firstly, what is meant by "randomly selected"? What was the selection process? Why not taking all patients in a limited time frame? Secondly, selection from patients "...suspected infection, and underwent serum PCT testing…", what does this mean?, What were the criteria for testing serum PCT? In the reviwer's view, there was an uncontrolled patient selection bias.

I added the sampling method in detail and revised the paragraph of Study Participants and Inclusion     Criteria as follows

Participants were included in the analysis if they had a febrile sense, suspected infection and underwent serum PCT testing based on the ED physician decision. We used systemic random sampling method. During study period, we selected first patient visiting emergency room after 9 am and 9 pm among the patients with suspected infection, and enrolled the patients if they underwent serum PCT testing. We excluded patients who were under 18 years, pregnant or had limited life expectancy due to chronic diseases. (Figure 1) (line: 74-79)

It is mentioned that the standard serum PCT test results are available within one hour, which is quite reasonable. Hence, what is the real time that was saved by the new whole blood test? Did this faster test result in any quicker decisions? Were the whole blood PCT tests blinded to the ED physicians?

I also wanted to investigate the advantages of POCT such as shortening the decision time as you pointed out. But we did not intervene in the process of patient care or any decision except whole blood procalicitonin testing. So, I couldn’t demonstrate advantage by actual data and only describe the theoretical advantages in discussion. I did not consider the blindness of whole blood PCT test because its result did not affect the clinical process.

I already mentioned this content in the limitation section. (182-184)

It remains unclear how the patients "without bacteremia" were defined. Were these 158 patients (199 total minus 41 BC positive?), although there may be another severe infection?? If so, what is the value of the high NPV? Not giving antibiotics to patients with severe pneumonia, but negative BC??

We defined the bacteremia as positive peripheral blood cultures in a patient with signs and symptoms of infection per the textbook definition. (line 89)

The number of patients underwent blood culture, sputum culture and urine culture tests was 182, 112 and 127, respectively. The number of positive blood culture was 41 in 182 patients underwent blood culture. (119-120)

I think NPV of procalcitonin is just one factors to be considered in giving antibiotics. I don’t want to make any opinion on this result. I'd like to leave the matter to the experts who read this article.

I revised the results of table based on the number of patients underwent each culture.

Linear regression is not the standard method to analyze a new test (whole blood PCT) compared with a gold standard (serum PCT). Please provide a Bland-Altman analysis plus plot!!

I added a Bland-Altman analysis plot and intraclass correlation coefficient value.

It is absolutely unclear why in Table 3 and Figures 2 and 3 (AUROC), the standard serum PCT is left out?? Direct comparison is one of the key questions!

I added data of standard serum PCT in table 3, figure 2 & 3. I revised the results and discussion according to the added data.

Minor Comments:

Abstract: What is an "accurate correlation"??? This is not a standard wording in statistics.

I changed the word “accurate” into “excellent”. (line 30)

References: Some of the papers referenced (especially in the Intro) are more than 20 years old.

I changed the old reference into new ones. (line 235, 239)

Statistics: How did the Authors test normal distribution?

An independent sample t-test was used for normally distributed continuous variables examined by the Kolmogorov-Smirnov normality test (line 100-101)

Page 3, ll. 115 to 117: Probably the AUROC for mortality is meant. Please add.

Yes, I ad

Round  2

Reviewer 1 Report

The revised manuscript was well received. The revision is satisfactory. Authors did significantly improvement to the manuscript  There are no further improvement need. However, there are minor spellings error need to be corrected. For line 62 and line 65, the word "object" shall change to "objective".

Thank you.

Reviewer 2 Report

all suggestions were followed